# Bone-Targeted Delivery of Novokinin as an Alternative Treatment Option for Rheumatoid Arthritis

**DOI:** 10.3390/pharmaceutics14081681

**Published:** 2022-08-12

**Authors:** Arina Ranjit, Sana Khajeh pour, Ali Aghazadeh-Habashi

**Affiliations:** College of Pharmacy, Idaho State University, Pocatello, ID 83209, USA

**Keywords:** novokinin, adjuvant arthritis, anti-inflammatory, renin-angiotensin system, arachidonic acid metabolite, bone-targeted drug delivery

## Abstract

Rheumatoid arthritis (RA) is an autoimmune inflammatory bone destructive disorder that is orchestrated by multiple systems in the body, including Renin-Angiotensin System (RAS) and arachidonic acid (ArA) pathway. Current therapeutic options are not highly effective and are associated with severe side effects, including cardiovascular complications. Therefore, new safe and effective disease modulators are seriously needed. In this study, we investigate the anti-inflammatory effects of a synthetic peptide, novokinin, through Angiotensin Type (II) receptor (AT2R). Peptide drugs like novokinin suffer from plasma instability and short half-life. Thus, we developed a novel bone targeting novokinin conjugate (Novo Conj). It uses the bone as a reservoir for sustained release and protection from systemic degradation, improving stability and enhancing pharmacological efficacy. We tested Novo Conj’s anti-inflammatory effects in adjuvant-induced arthritis (AIA) rat model to prove our hypothesis by measuring various RAS and ArA pathway components. We observed that inflammation causes a significant imbalance in cardioprotective RAS components like ACE2, AT2R, and Ang 1-7 and increases the ArA inflammatory metabolites like hydroxyeicosatetraenoic acids (HETEs). Treatment with novokinin or Novo Conj restores balance in the RAS and favors the production of different epoxyeicosatrienoic acids (EETs), which are anti-inflammatory mediators. This study demonstrated that the bone-targeted delivery improved the stability and enhanced the anti-inflammatory effects of the parent peptide novokinin in AIA. These observations offer an efficacious alternative therapy for managing RA.

## 1. Introduction

About 1% of the world’s population is affected by rheumatoid arthritis (RA), an autoimmune bone destructive disorder [1]. It is a chronic inflammatory disease characterized by synovitis, an incursion of inflammatory cells, and progressive joint destruction. The inflammatory response in the RA involves multisystem effects in the cardiovascular, pulmonary, and psychological besides skeletal disorders [2]. Compared with the general population, patients with RA have a lower quality of life and a higher risk of early morbidity due to gradual disability and systemic complications leading to increased socioeconomic burden [3]. There is no cure for RA, but treatment with nonsteroidal anti-inflammatory drugs (NSAIDs), corticosteroids, disease-modifying antirheumatic drugs (DMARDs), and biologics put patients in remission of symptoms. These agents are associated with mild to severe side effects, ranging from reduced patient compliance to life-threatening, including stomach irritation, cardiovascular problems, kidney damage, liver damage, severe lung infections, and cancer [4]. Therefore, safe and effective alternative treatment options are needed.

Renin-angiotensin system (RAS) is a circulating endocrine system primarily responsible for blood pressure and fluid–electrolyte balance [5,6]. With the advent of evidence of local functional RAS system, its implications have been found beyond cardiovascular disorders such that components of classical arm RAS like angiotensin II (Ang II), angiotensin-converting enzyme (ACE), and angiotensin type I receptor (AT1R) are upregulated in synovial tissue of RA specimens and found to be responsible for the exacerbation of RA [7]. In contrast, the components of the alternative arm of the RAS like angiotensin type II receptor (AT2R) and angiotensin-converting enzyme 2 (ACE2)/angiotensin 1-7 (Ang 1-7)/mas receptor (MasR) axis (ACE2/Ang 1-7/MasR) are reported to have a role in the resolution of inflammation [8]. There have been indications that RAS components directly influence the pathogenesis of RA through the modulation of inflammatory mediators and angiogenesis, which points out its suitability for the treatment and management of RA [9]. Although AT2R exists as a lesser-known axis, it has stood out as a potential alternative pathway due to its upregulation in pathological conditions. It can be channelized to functionally antagonize AT1R, a primary effector receptor in the RAS [5].

The cytochrome P450 (CYP)-mediated arachidonic acid (ArA) metabolism poses as another essential precursor of various bioactive molecules in regulating inflammation, thus participating in the pathogenesis of RA [10]. The eicosanoids are generated by the action of enzymes like CYP4A and CYP4F on ArA, which by ω- hydroxylation producing hydroxyeicosatetraenoic acids (HETEs), act as pro-inflammatory mediators. In contrast, its epoxidation-producing epoxyeicosatrienoic acids (EETs) act as anti-inflammatory mediators. Correspondingly, dihydroxyeicosatrienoic acids (DiHTs), metabolites of EET, by soluble epoxide hydrolase, also act as anti-inflammatory molecules but are less potent than their parent molecule [11,12]. Along with this, there have been ample corroborations that the interplay of RAS and CYP-mediated ArA pathway can play a significant role in the manifestations of RA [13,14]. The AT2R activation has also been reported to favor EETs production and inhibit intracellular signaling of pro-inflammatory cytokines [15].

Novokinin, a six amino acid synthetic peptide (Arg-Pro-Lys-Leu -Pro-Trp) (MW = 796), first derived from the chymotryptic digest of ovalbumin, is known to mediate its biological effect as an AT2R agonist [16]. It is reported to have anti-hypertensive and vasorelaxant activity in the mesenteric artery isolated from spontaneously hypertensive rats [17]. Its therapeutic effects expanded to anorexigenic [18], gastroprotective [19], anti-inflammatory [20], and cyclooxygenase (COX)-inhibiting effects [18]. Here, we are reporting, for the first time, the anti-inflammatory effects of novokinin in a rat model of adjuvant-induced arthritis (AIA). Despite its multiple therapeutic effects, novokinin, like other peptide drugs, faces major challenges in drug development as they are cleared rapidly from the body. They have poor oral bioavailability as gastrointestinal enzymes easily degrade them. They also have a short half-life and are unstable due to poor absorption, distribution, metabolism, and excretion. All of which contribute to the low permeability, metabolic instability, and short residence time of the peptide drugs in the body [21].

Hence, a suitable drug delivery system with an appropriate target and protection from metabolic degradation is needed for the peptides. This approach should increase drug concentration at the desired site with lower systemic toxicity. Bone-targeted drug delivery has emerged as a rational approach among the various peptide delivery tactics. It targets the skeletal bone, and a specific conjugation with a polyethylene spacer mediates such capability. Several peptides like calcitonin [22], bovine serum albumin [23], and parathyroid hormone [24] have been previously conjugated with bisphosphonate (BP), which showed superior therapeutic activity compared to their parent compound. We propose the bone targeting approach by conjugating novokinin with a polyethylene glycol (PEG) linker and the bone-targeting moiety (BP) to give a novokinin conjugate (Novo Conj). Up on administration, the BP moiety attaches to the bone, which acts as a reservoir. The PEG linker protects novokinin from degradation, but over time the drugs get released and ensure the sustained release of active peptide. We hypothesized that Novo Conj would have superior anti-inflammatory effects mediated through AT2R in the rat model of AIA compared to its parent compound novokinin.

To study the pathogenic processes of arthritis, rodent models of RA are valuable tools. Among these animal models, the AIA has been widely used to investigate cardiovascular complications associated with RA and study the efficacy of anti-arthritic agents [25,26].

## 2. Materials and Methods

### 2.1. Peptide Synthesis, Conjugation, and Characterization of Novo Conj

Novokinin was synthesized using a standard Fmoc-mediated solid-phase peptide synthesis (SPPS) using the Aapptec Focus Xi peptide synthesizer (Louisville, KY, USA). All Wang resin, Fmoc-protected amino acids, solvents, and reagents needed for peptide synthesis were purchased from Aapptec (Louisville, KY, USA). The chain elongation was performed by removing the Fmoc group from the resin-attached amino acid. Next, Fmoc-protected amino acid (based on the sequence) was added until the entire peptide was synthesized. After the completion of synthesis, the final Fmoc group was removed and cleaved from the resin. After purification of the novokinin, it was coupled with Maleimidopropionyl-PEG NHS Ester (Polypure, Oslo, Norway) in an equimolar concentration for an hour to form an intermediate compound. A 10× solution of thiol-BP (Surfactis Technologies, Angers, France) was reacted with the intermediate compound for another hour. After completion of coupling, the conjugate was dialyzed to separate it from unreacted reagents.

The resultant peptide and conjugate were characterized by Agilent 6545 LC/Q-TOF. A US patent application for this innovative Novo Conj synthesis and pharmacological effect was filed. For the animal study, a batch of Novo Conj was synthesized by AnaSpec (CA, USA) and used after structural and purity confirmation.

### 2.2. In Vitro Hydroxyapatite Study

The bone-targeting ability of binding of Novo Conj was evaluated by in-vitro hydroxyapatite (HA) binding affinity test according to our previously published protocol [27]. Briefly, 20 µg of novokinin or an equivalent amount of Novo Conj was mixed with 5 mg of HA powder in 750 µL of various buffers like double-distilled water (D.D H_2_O), 10 mM PBS (pH = 7.4), and acetate buffer (pH = 4). An equivalent amount of novokinin and Novo Conj were mixed in corresponding buffers without HA and were used as a control. All the mixtures were shaken gently at room for 1 h and then centrifuged at 10,000× *g* for 5 min. The supernatant was collected and assayed for unbound drug using a fluorescence spectrometer at λE_x_ 215 nm and λE_m_ 305 nm (Varioskan Lux, Thermo Scientific, MA, USA). The percentage of HA binding study was calculated as (intensity of control-intensity of supernatants/intensity of control × 100%). All the experiments were run in triplicate.

### 2.3. Animal Study

The study protocol was in accordance with the approved protocols of the Idaho State University’s Institutional Animal Care and Use Committee (protocol #772, 09/22/2021). Adult healthy male Sprague–Dawley (SD) rats (230–250 g) were obtained from the Health Sciences Laboratory Animal Services. They had free access to food and water and were housed under standard temperature, ventilation, and hygienic conditions with a 12 h light and dark cycle. Animals were acclimatized for 72 h before starting the experiment.

#### 2.3.1. Induction and Assessment of Experimental AIA

Experimental AIA in healthy male SD rats was induced by administering 0.2 mL of *Mycobacterium butyricum* (Difco Laboratories, Detroit, Michigan) solution (50mg/mL in squalene) on day 0 by the tail base injection as previously described [28]. Arthritic rats were divided into 3 groups and were given saline (Arthritic, *n* = 6), novokinin (Arthritic + novokinin, *n* = 5), and Novo Conj (Arthritic + Novo Conj, *n* = 5), and a group of healthy rats that received saline (Control, *n* = 6). The treatment groups received an equivalent dose of 400 µg/kg of novokinin solution in saline thrice a week (S.C) for 2 weeks after the emergence of arthritis. Subsequently, body weight using a regular balance and joint and paw diameter changes with a caliper with 25 µm sensitivity (Mitutoyo Canada Inc., Toronto, ON, Canada) were measured periodically. The progression of AIA was monitored daily, and the arthritis index (AI) was calculated using a macroscopic scoring system as described before [28]. Briefly, each hind paw was rated on a scale of 0–4: 0—no virtual sign of arthritis; 1—involvement of single joint; 2—involvement of greater than 1 joint and/or ankle; 3—involvement of several joints and ankle with moderate swelling; and 4—involvement of several joints and ankle with severe swelling. Each forepaw was rated on a scale of 0–3: 0—no virtual sign of arthritis; 1—involvement of single joint; 2—involvement of greater than 1 joint and/or wrist; and 3—involvement of wrist and joints with moderate to severe swelling. The scores from each paw were added for the calculation of AI. A score of ≥5 was considered a significant disease emergence, with 14 as the highest possible index that could be observed in the AIA rat. The inclusion criteria for the rats to receive treatment was having at least one paw or joint swelling. Any rats that developed AI ≥ 5 were excluded and euthanized as a humane masseur of animal use and handling. All animals were euthanized at the end of the experiment, and serum, plasma, and heart tissue were harvested. Serum samples were analyzed for total nitrate and nitrite (nitric oxide, NO). RAS enzymes and protein through qPCR and Western blotting, and the Ang 1-7, Ang II, and ArA metabolites were analyzed by LC-MS/MS methods.

#### 2.3.2. Assessment of Nitric Oxide

A nitrite/nitrate assay kit (Sigma-Aldrich, St. Louis, MO, USA) was used to quantify NO metabolites in serum according to the experimental protocol provided by the kit manufacturer (Molecular Probes, Invitrogen detection technology, Eugene, OR, USA). NO levels in serum are short-lived and were thus converted to its stable metabolites nitrate (NO_3_) and nitrite (NO_2_) using the Griess reaction and measured by colorimetric detection. Briefly, samples were incubated with nitrate reductase to reduce nitrate to nitrite. Griess reagent was added, and absorbance intensity was measured at 570 nm to evaluate total nitrite levels (Varioskan Lux, Thermo Scientific, MA, USA). Background absorbance was calculated for each sample and subtracted from total values. Based on the kit’s manufacture pamphlet, the lower limit of quantification (LLOQ) was about 82.5 ng/µL, which was way below the levels we detected in the serum samples.

#### 2.3.3. Quantitative Real-Time PCR (qRT-PCR)

Total RNA was extracted from rat cardiac tissue samples using Quick-RNA^(TM) Miniprep Plus kit (Zymo Research). Five hundred nanograms of RNA was transcribed into cDNA using qScript cDNA super mix (P/N 84034, Quanta Bio, Beverly, MA, USA). Twenty nanograms of cDNA was used for the qRT-PCR. The cDNA samples in a triplicate were used as templates for qRT-PCR using the PerfectaSYBR Green fast master mix (P/N 84069, Quanta Bio, Beverly, MA, USA) in a Master cycler Epgradient S (Eppendorf, Enfield, CT, USA). The expression of target genes was normalized by GAPDH in the same cDNA sample (Table 1). The relative expression of target genes was determined using the 2^−ΔΔCt^ method, where ΔCT_(control)_ = CT_(Target gene)_ − CT_(GAPDH),_ ΔCT_(test)_ = CT_(Target gene)_ − CT_(GAPDH)_. The mRNA expressions were normalized to the house GAPDH and presented as fold change compared with control. Three samples were used per experimental group. Triplicates of qPCR experiments were carried out for statistical analysis.

#### 2.3.4. Western Blotting

Rat cardiac tissue (25 mg) was homogenized and mixed in 1500 µL of RIPA buffer with a protease inhibitor tablet (1 tablet per 10 mL) (P/N 11697498001, Sigma Aldrich, St. Louis, MO, USA). The supernatants were taken after centrifugation for protein quantification with the Qubit Protein Reagent. The same amount of protein (128 µg) per lane was separated electrophoretically by Tris-Glycine (4–12%) Gel and transferred to a PVDF membrane. The membranes were incubated with a primary Ab, ACE (rabbit monoclonal EPR22971-247 to ACE1, ab25422, Abcam1:1000), and ACE2 (rabbit monoclonal EPR4435(2) to ACE2, ab108252, Abcam1:1000), AT1R (Rabbit monoclonal EPR3873 to AT1R, ab124734, Abcam1:1000), AT2R (rabbit monoclonal EPR3876 to AT2R, ab92445, Abcam1:1000), and β-actin (mouse monoclonal AC-15 to β- actin, ab6276, Abcam1:1000) overnight in 4 °C. After washing four times for 10 min in TBS containing 0.1% Tween 20, the membranes were incubated with rabbit anti-mouse IgG with HRP secondary Ab (NBP1-73435, Novus 1:10,000) for 1 h at room temperature. Membranes were washed, incubated with Radiance Q Chemiluminescent substrate (Azure Biosystems, Dublin, CA, USA), and imaged by Azure biosystems c600 for 5 s^−1^ min. The density of a specific band was quantified using ImageJ software.

#### 2.3.5. Ang 1-7 and Ang II Level by LC-MS/MS

##### Chemicals

Ang 1-7 (Anaspec, AS-61039) and Ang II (Anaspec, AS-20633) were purchased from Anaspec Inc. (Fremont.CA, USA). Correspondingly, [asn1, Val5]-angiotensin II) used as Internal Standard (IS) (Sigma-Aldrich A6402-1MG) was bought from Sigma Aldrich (St. Louis, MO, USA). C18 column (SepPak WAT020805) was bought from Waters (Milford, MA, USA). LC-MS grade water, acetonitrile, and formic acid were purchased from Fisher Scientific (Fair Lawn, NJ, USA).

##### Instrumentation

All LC-MS/MS experiments were carried out in an AB SCIEXQTRAP^®^ 5500 mass spectrometer (Foster City, CA, USA) in tandem with a Nexera HPLC system from Shimadzu Corporation (Columbia, MD, USA). The HPLC system consists of a Sil-30AC autosampler, LC-30AD pumps, a CBM -30 A controller, a DGU-20A5R degasser, and a CTO-20A column oven. Analyst and Multiquanta software were used for data acquisition and quantitation.

##### Chromatographic and Mass Spectrometric Condition

Rat plasma samples were taken and analyzed for Ang 1-7 and Ang II levels by the published LC-MS/MS method [34]. The method was applied for plasma sample analysis after minor modification and validation. Briefly, Ang peptides were separated by Synergi RP (2 × 100 mm) column with a 2.5 µm particle size (Phenomenex, CA, USA) at ambient temperature. The mobile phase consisted of 0.1% formic acid in water (A) and ACN (B). The gradient time program started from 5% ACN to 30% ACN over 4 min, and composition was kept constant for 4 to 8 min, lowered to 5% ACN to 9 min, and ran for 10 min. The flow rate was 0.3 mL/min, and the injection volume was 30 µL. Electrospray ionization was used, and analytes were detected using multiple reaction monitoring (MRM) in the positive mode. The optimized source/gas parameters were as follows: curtain gas, 30; collision gas, medium; ion spray voltage, 5500 V; temperature, 300 °C; ion source gas 1 (nebulizer gas), 20 psi; and ion source gas 2 (turbo gas), 25 psi. LC-MS analysis was performed with the single ion recording (SIR) mode, in which the *m*/*z* 300.5, 349.6, and 516.6 were used for Ang 1-7, Ang II, and IS, respectively. LC-MS/MS was performed with MRM transitions of *m*/*z* 300.6→136 (Ang 1-7), *m*/*z* 349.6→136 (Ang II), and *m*/*z* 516→769.4 (IS).

##### Plasma Sample Preparation

An aliquot of 200 µL plasma, 100 µL of IS (100 ng/mL), and formic acid was added to the final concentration of 0.5% and mixed well. Solid-phase extraction (SPE) was carried out in Agilent PPE manifold 48 processors. The sample solutions were applied to the C18 column that was preconditioned with 2 mL of ethanol and 2 mL of deionized water, respectively. After loading the sample, the cartridge was washed with 2 mL of deionized water. Water was left to dry for 3 min by turning on a high flow of nitrogen gas. Then, Ang peptides were eluted from the cartridge using 2 mL methanol containing 5% formic acid, collected, and dried under the stream of nitrogen. Finally, the dried sample was reconstituted in water and transferred to a sample vial for LC-MS/MS analysis.

The calibration curves were constructed over the range of 0.31–5 ng/mL for Ang 1-7 and 200–3200 pg/mL for Ang II for LC-MS/MS injection. The Ang peptides’ concentrations were determined by comparison using the calibration curves. The precision and accuracy were evaluated by analyzing spiked plasma samples in triplicate at the following concentrations presented in Table 2. Accuracy was calculated using the following equation: Accuracy (%) = [(Mean observed measured concentrations/spiked concentrations)] × 100. The LLOQ was 0.15 ng/mL and 100 pg/mL for Ang 1-7 and Ang II, respectively.

#### 2.3.6. ArA Metabolites Analysis by LC-MS/MS

##### Chemicals

The reference standards were purchased from Cayman Chemical Company (Ann Arbor, MI): 19-(R)-hydroxyeicosatetraenoic acid (19-HETE) (P/N,10007767), 20-hydroxyeicosatetraenoic acid (20-HETE) (P/N,90030), (±)-5,6-epoxyeicosatrienoic acid (5,6-EET)(P/N,50211), (±)8,9-epoxyeicosatrienoic acid (8,9-EET) (P/N,50351), (±)11,12-epoxyeicosatrienoic acid (11,12-EET)(P/N,50511), (±)14,15-epoxyeicosatrienoic acid (14,15-EET)(P/N,50651), (±)-5,6-dihydroxyeicosatrienoic acid (5,6-DiHT)(P/N, 51211), (±)8,9-dihydroxyeicosatrienoic acid (8,9-DiHT)(P/N,51351), (±)11,12-dihydroxyeicosatrienoic acid (11,12-DiHT)(P/N,51511), and (±)14,15-dihydroxyeicosatrienoic acid (14,15-DiHT)(P/N,51651). Additionally, the following deuterated internal standards (IS) were also obtained from Cayman: 8,9-EET-d11 (deuterium atoms at the 16,16,17,17,18,18,19,19,20,20, and 20 positions; isotopic purity of ≥99%).

##### Instrumentation

Similar instrumentation to Ang peptide analysis was used for the ArA metabolite assay.

##### Chromatographic and Mass Spectrometric Conditions

The experimental protocol and assay condition were followed with slight optimization to analyze ArA metabolites as previously described [35]. Briefly, eicosanoids were separated using a Synergi RP (2 × 100 mm) column with a 2.5 µm particle size (Phenomenex, CA, USA) at ambient temperature. The mobile phase consisted of 0.1% formic acid in water (A) and CAN (B). The mobile phase gradient time program started from 5% ACN to 20% ACN over 2 min, then increased to 55% ACN, kept constant for 2.5 to 6 min, increased to 100% ACN to 8 min, and ran until 9 min. Then, it was subsequently reduced down to 5% of ACN over a 10.5 min total run. The flow rate was 0.3 mL/min, and the injection volume was 10 µL. The mass spectrometric condition consists of a triple quadrupole that is used to monitor their *m*/*z* transition by Analyst software. Electrospray ionization was used, and analytes were detected using MRM in the negative mode. The optimized source/gas parameters were as follows: curtain gas, 20 psi; collision gas, medium; ion spray voltage, −4500 V; temperature, 400 °C; ion source gas 1 (nebulizer gas), 20 psi; and ion source gas 2 (turbo gas), 25 psi. LC-MS/MS analysis was performed with MRM transitions for ArA metabolites according to the parameters listed in Table 3.

##### Plasma Sample Preparation

An aliquot of 300 µL plasma sample was mixed with 100 µL of 8,9-EET d11 (IS, 100 ng/mL), and 2 µL of FA was added. Then, samples were vortex mixed and extracted with 500 µL of ethyl acetate twice. Each time sample was centrifuged at 15,000× g, 4 °C, for 10 min. Three hundred microliters of the supernatant layer was taken after the first addition of ethyl acetate, and 500 µL of the supernatant layer was taken after the second addition of ethyl acetate. After mixing the separated organic phases, it was dried under nitrogen gas and reconstituted in methanol for LC/MS-MS analysis.

The calibration curves were constructed over the range of 0.625–160 ng/mL for each ArA metabolite. The concentrations of each metabolite in the plasma samples were determined using the corresponding calibration curve. The LLOQ, precision, and accuracy were evaluated by analyzing spiked plasma samples in triplicate at the following concentrations listed in Table 4.

### 2.4. Statistical Analysis

Data were analyzed by a standard computer program, GraphPad Prism Software PC software, version 9.3.1, Statistical Package for Social Sciences (SPSS) version 26 for Windows (SPSS Inc., Chicago, IL, USA) or Minitab software version 20 (Minitab LLC, Chicago, IL, USA) and are expressed as mean ± standard deviation (Mean ± SD) of at least three independent experiments. Data were analyzed for normal distribution and homogeneity of variance before proceeding with the parametric statistical tests. The student’s t-test analyzed in-vitro binding affinity between two groups. One-way ANOVA analyzed differences between mean values of multiple groups with Tukey’s test for post-hoc comparisons. To compare the % weight gain over time between treatment groups, a two-way ANOVA with post hoc Tukey’s test was used. AI over time between treatment groups was tested by the non-parametric Kruskal–Wallis analysis of variance. Mann–Whitney U-test was used to compare the significant difference between groups by SPSS. The linear correlation of ArA metabolites and RAS components was analyzed using Graphpad Prism software. The orthogonal regression analysis was conducted to account for errors in observations on both the x- and the y- axis and find a line of best fit using Minitab software. *p*-values of less than 0.05 were considered statistically significant. The data labeled with different letters (a, b, c, or d) in tables or figures indicate a statistical difference between groups where *p* < 0.05.

## 3. Results

### 3.1. Synthesis and Characterization of Novokinin and Novo Conj

Novokinin and Novo Conj were successfully synthesized by FMOC SPPS and in-solution methods, respectively. The characterization of synthesized novokinin and Novo Conj was performed on LC/Q-TOF. Extracted ion chromatogram (EIC) depicted in Figure 1A shows 796 and 398 ions, which correspond to M + 1 and M + 2, respectively, for the novokinin with a molecular weight of 796 amu. Similarly, in Figure 1B, the EIC shows that 1274, 850, and 637.82 ions correspond to M + 2, M + 3, and M + 4 for Novo Conj with a molecular weight of 2546 amu.

### 3.2. In Vitro Hydroxyapatite Binding Study

The bone mineral affinity of Novo Conj was compared with novokinin using in-vitro HA binding affinity test. As shown in Figure 2, Novo Conj had 16.74 ± 1.0%, 6.89 ± 1.8%, and 7.65 ± 1.13% binding capacity, whereas in the case of novokinin it was 0.89 ± 0.29%, 1.52 ± 1.48%, and 2.14 ± 1.1% in D.D H20, acetate buffer, and PBS 10 mM respectively. This result shows that the bone mineral affinity of Novo Conj was significantly higher in each medium compared to novokinin.

### 3.3. Animal Study

#### 3.3.1. Effect of Novokinin and Novo Conj on Body Weight Gain and Clinical Symptoms of AIA

Adjuvant-induced arthritis was noticeable in 100% of animals after 10–12 days of adjuvant inoculation. It caused a significant reduction in body weight gain and increased the AI and paw and joint diameter in 24 days (Figure 3 and Table 5). The treatment with novokinin and Novo Conj restores the body weight over time. The effect of Novo Conj on the improvement of bodyweight was significantly better at the end of the treatment than in its counterpart, novokinin-treated rats. Similarly, a higher AI score was seen in arthritic rats when clinical symptoms of AIA, like the number of paws and joints involved and their redness and swelling, were used to calculate the AI (Figure 3B). A marked reduction in AI scores was observed in Novo Conj and novokinin treated rats at the end of the regimen, which was significantly different from the arthritic group. As reported in Table 5, healthy rats have a standard percentage increase in paw and joint diameter due to their growth in body weight compared to day 0. However, the swelling due to inflammation caused an apparent percent increase in the paws and joint diameter in the arthritic rats compared to day 0. The swelling rate was diminished after the drug treatment, but Novo Conj treatment resulted in a better resolution of inflammation and reduced swelling.

#### 3.3.2. Novokinin and Novo Conj Reduce the Elevated Level of NO

NO is a marker of nitrosative stress, and it directly participates in the pathogenesis of RA, as reported in several studies [36,37]. So, we measured serum total NO as an indication of resolution of inflammation by the treatment of Novo Conj. The treatment with novokinin or Novo Conj reduces elevated NO levels with more significant effects seen in Novo Conj-treated rats, as shown in Figure 4.

#### 3.3.3. Effect of Novokinin and Novo Conj on mRNA Expression of RAS Components

The mRNA expression of primary RAS components was analyzed in cardiac tissues of rats belonging to different treatment groups. The mRNA expression of ACE2 has downregulated while that of ACE and AT1R is upregulated in inflamed rats compared to control and other treatment groups. This effect was reversed after novokinin or Novo Conj treatment, and there was an enhancement in ACE2 expression and a decline in elevated ACE and AT1R (data not shown). Figure 5A shows the ACE2/ACE fold increase ratio, normalized and compared to control in the cardiac tissues. The treatment with Novo Conj improved the ACE2/ACE gene expression ratio significantly better than its parent compound. There was a trend in the increase of mRNA expression of AT2R in the arthritic rats compared to control, and with the treatment, it was further increased (data not shown). However, as shown in Figure 5B, the AT2R/AT1R ratio fold was lower in the arthritic group compared to the control group. The treatment with Novo Conj upregulated the AT2R/AT1R ratio fold significantly higher than the control, arthritic, and arthritic+novokinin-treated groups.

#### 3.3.4. Effect of Novokinin and Novo Conj on Protein Expression of RAS Components

The protein expression of RAS components was investigated in cardiac tissue. The enzyme and receptor associated with inflammatory axis ACE and AT1R were upregulated in arthritic rats. The drug treatment lowered their expression with a significant change seen in Novo Conj-treated rats than in the novokinin-treated group (Figure 6A). In addition, a reverse effect was seen in the expression of an enzyme belonging to the anti-inflammatory axis ACE2 such that the ACE2/ACE ratio was higher in the Novo Conj group and comparable to the control. Similar to its gene expression, the protein expression of AT2R was upregulated in arthritic rats, and the treatment with drugs further increased their expression (data not shown). However, as shown in Figure 6B, the AT2R/AT1R protein expression ratio was significantly lower in the arthritic rats than in the control group. The treatment with novokinin restores it to a comparable level as the control group, while Novo Conj treatment increases significantly compared to other treatment groups.

#### 3.3.5. Effect of Novokinin and Novo Conj in Ang 1-7 and Ang II Plasma Levels

Besides having effects on the RAS enzymes and receptors level, treatment with novokinin and Novo Conj impacted the RAS peptides. AIA reduces the Ang 1-7 plasma level in the arthritic group, and the treatment helps to restore its level. The Novo Conj showed significantly better effects than novokinin in restoring depleted Ang 1-7 levels in arthritic rats, as shown in Figure 7A. In addition, there was an increment in the plasma level of Ang II, which resulted in a dramatic reduction of the Ang1-7/Ang II ratio in the arthritic rats (Figure 7B,C). The administration of novokinin and Novo Conj restored the peptides ratio by lowering Ang II and increasing the Ang 1-7 levels to comparable values of the control group, mainly in the case of Novo Conj (Figure 7).

#### 3.3.6. Effect of Novokinin and Novo Conj on ArA Metabolites Levels and Their Correlation with RAS Components

The treatment with Novokinin or Novo Conj lowered the upregulated total-HETE level (T-HETEs) in the AIA rats shown in Figure 8A and presented with a positive impact on increasing total-EETs level (T-EETs) and total-DiHTs (T-DiHTs) in the plasma at the level of control as reported in Figure 8B,C. Consequently, the ratio of T-EETs/T-HETEs was more significantly upregulated in the plasma of Novo Conj-treated rats than the novokinin treated one, which stood out as better treatment options when ArA metabolites ratios were compared to the other treatment groups (Figure 8D).

We also observed significant linear correlations between different ArA metabolites and RAS peptide levels in the plasma (Table 6). Strong and positive correlations existed between some individual EETs, DiTHs, T-EETs, T-DiTHs, T-EETs/T-HETEs, and T-DiHTs/T-HETEs with Ang 1-7 plasma levels. At the same time, those metabolites were inversely correlated with Ang II plasma levels. Accordingly, we observed a strong positive correlation between individual HETEs and T-HETEs with Ang II levels. In contrast, the sign of correlations was inversed in the case of Ang 1-7.

To account for errors in observations on the x- and y- axis (RAS peptides and ArA metabolites) and find the line of the best fit, we conducted orthogonal regression analysis. The analysis outputs are presented in Appendix A and Appendix A. The best-fitted least-square and orthogonal regression lines between RAS components with ArA metabolites of Total-EETs, Total-HETE, and Total-DiHT (Appendix A) indicate that both regression lines are the same in the case of Ang 1-7, and they are also well-aligned with each other in the case of Ang II. The normal probability plots (Appendix A) display that the data sets are approximately normally distributed and no outliers were identified in the data sets. The residuals of calculated values of RAS components are plotted against the experimental values (Appendix A). As illustrated, the propagation of the residuals on both sides of the zero line indicate that no symmetric error exists in the development of the regression models.

## 4. Discussion

Several biologically active peptides boast multiple advantages in affinity and selectivity over small drug molecules. Still, few have moved forward in the drug development process due to the short half-life and instability [38]. Several approaches like modification of amino acids and cyclization have been utilized to address the issue as they offer resistance against enzymatic degradation. Often these approaches lead to an alteration in the efficacy of peptides and require more complex remodeling to restore biological activity [39]. PEGylation of peptides provides appropriate alternatives for these approaches. It not only retains the original structure of the peptide but also increases the hydrophilicity, decreases renal clearance, reduces accessibility for proteolytic enzymes, and diminishes immunogenicity and antigenicity [40]. On the other hand, the bone-targeting approach that attaches BP to the peptide–PEG complex helps to increase stability by utilizing bone as a drug reservoir. This conjugation improved pharmacokinetics and bone tissue accumulation of parathyroid and Ang 1-7 peptides [24,27]. In our current study, we are reporting, for the first time, that the modifications of novokinin by conjugation with PEG and BP show similar effects. We successfully synthesized novokinin and did subsequent PEGylation and conjugation with BP, confirmed by mass spectrometric detection in Figure 1. We opted for SPPS for Novokinin synthesis as it provides benefits over solution-based syntheses such as easy purification, shorter time for synthesis, and convenient modifications, and it yielded a pure compound (>95%). However, in-solution synthesis was more feasible in the case of the experimental synthesis of Novo Conj. The in-vitro study of Novo Conj bone mineral affinity (Figure 2) proves that Novo Conj has a high affinity for binding to HA of the bone, similar to Ang 1-7 conjugate [27].

Although collagen-induced arthritis is the most relevant model, the literature indicates that AIA is appropriate for studying cardiovascular and other complications associated with human RA [25,26]. As we were interested in investigating the cardiovascular protection of Novo Conj through RAS and ArA pathways, we chose the AIA model for our study. In the present work, we observed almost 100% incidence of AIA through a single administration of *Mycobacterium butyricum*, as reported previously [28]. The role of the AT2R in the resolution of inflammation in the RA rat model has also been studied by AT2R agonists like CGP42112 [41]. To explore the anti-inflammatory spectrum of AT2R, we tested the effect of novokinin, a synthetic peptide AT2R agonist, and its conjugate, Novo Conj, modulating the experimental AIA. The dose of novokinin was selected based on previous studies to test its other biological activities. Novokinin has been used in a wide range of 0.1 mg/kg intravenously to 30–100 mg/kg orally [42,43]. In this pilot study, we chose a 0.4 mg/kg dose to be delivered subcutaneously every other day (thrice weekly for 2 weeks) to test its anti-inflammatory effect. The animals tolerated both novokinin and Novo Conj-administered dose well. Later, we conduct a dose-response and toxicology study on a wide range to find a more effective and safe dose.

In concert with the earlier reports, weight loss and swelling of joints and paws were major hallmarks of successful induction of AA [28,41]. Without any treatment, these symptoms tend to get worse in AIA rats, but the intervention with drugs helps recover by restoring the weight loss and decreasing swelling of joints and paws. The subsequent reduction in AI score was realized in Novo Conj-treated rats with significant improvement in resolution of inflammation, as shown in Figure 3. Similarly, NO has been found to play an essential role in the pathogenesis of inflammation, as demonstrated by extensive studies [44,45]. The administration of Novo Conj prevents the rise in the serum nitrite concentration, the stable form of NO metabolite, further validating its anti-inflammatory role shown in Figure 4.

Apart from the physical symptoms, here in our study, we analyzed the molecular mechanism of RA through the lens of RAS. It is a known fact that an interplay between RAS and the inflammatory axis controls the outcome of RA [8]. The overactivation of the classical arm of RAS, i.e., ACE/Ang II/AT1R, is known to stimulate the pro-inflammatory cytokines, exacerbate angiogenesis, and promote osteopenia, thus leading to the gradual destruction of the joints and increased deformation and dysfunction [46,47]. In contrast, stimulation of the alternative arm of RAS, i.e., AT2R and ACE2/Ang 1-7/MasR axis, can lead to inhibition of oxidation, promotion of osteogenesis, and downregulation of pro-inflammatory mediators [41,48]. In this study, we demonstrated that novokinin and Novo Conj utilize the activation of an alternative RAS axis to resolve inflammation. We saw an increase in ACE2/ACE ratio and AT2R/AT1R ratio in the mRNA and protein expression in cardiac tissues obtained from treated AA rats, as shown in Figure 5 and Figure 6. Our results further corroborate that RA is involved in the pathogenesis of cardiovascular diseases [49].

Our data suggest that the AT2R stimulation is associated with ACE2 activation. This event is in agreement with the outcome of several studies that reported that the activation of AT2R increases ACE2 activity, which was seen in diabetic nephropathy [50], obesity-related hypertension [51], and inflammation [52]. In our study, we found a significant increase in Ang 1-7/Ang II ratio as Ang 1-7 and Ang II levels were increased and decreased, respectively, in the plasma of novokinin or Novo Conj-treated AIA rats. As ACE protein levels did not change by drug treatments, the rise in Ang 1-7 can be attributed to two factors (i) direct conversion of Ang II to Ang 1-7 through the enhanced ACE2 expression and (ii) indirect conversion by ACE2 by the promotion of Ang I metabolism to Ang 1-9, which in turn yields Ang 1-7 [53]. This fact led to the rationale that ACE2 activation led to a shift in the power balance between Ang 1-7 and Ang II in favor of the RAS protective arm. Thus, novokinin and, more efficiently, the Novo Conj reduce inflammation by promoting alternative RAS axis by restoring the balance.

Besides creating an imbalance in the RAS, inflammation also causes alterations in CYP expression, which controls one of the critical pathways for the metabolism of ArA [8]. For instance, there is increased ω-hydroxylase expression, encoded predominately by the CYP4A and CYP4F, which gives rise to HETE metabolites from ArA. The 20-HETE enhances the production of inflammatory cytokines/chemokines (IL-8, IL-13, IL-4, and prostaglandin E2) and exacerbates RA [54]. At the same time, there is decreased epoxygenase encoded by CYP2C and CYP2J, leading to a lower level of EETs and DiHTs, suppressing their anti-inflammatory activity [55]. Our study also found an association between inflammation and CYP-derived eicosanoids as arthritic rats have higher T-HETEs but lower T-EETs and T-DiHTs concentrations. Nevertheless, we saw an increase in T-EETs and T-DiHTs and a decrease in T-HETEs concentration in drug-treated rats with Novo Conj treatment able to bring to the control level (Figure 8). The rise in EETs and its comparatively low potent product DiHTs, counteract inflammation by reducing cytokines like TNF-α as well as inhibiting the transcription of the nuclear factor-kappa (NF-κB) translocation, thereby blocking leukocyte adhesion. In addition, their rise also downregulates different enzymes, such as lipoxygenase-5 and cyclooxygenase 2, as well as the inducible nitric oxide synthase (iNOS) that together are held responsible for the aggravation of inflammation [56].

Our findings show a strong correlation (with opposite signs) between some individual EEts, HETEs, T-EETs, and T-HETE with Ang 1-7 or Ang II levels (Table 6). These correlations were confirmed by orthogonal regression analysis data and the line best fit graphs (Appendix A and Appendix A). This interconnection hints that the modulators of both RAS and CYP-derived eicosanoids work in tandem to resolve or worsen inflammation. We also found a mild positive correlation between DiHTs and Ang 1-7 and an inverse negative correlation between DiHTs and Ang II levels, affirming that they are low potent anti-inflammatory mediators (Table 6). The AT2R stimulation through Novo Conj increases EETs and Ang 1-7 levels, thus promoting recovery and healing in AIA rats. Agonist novokinin falls short in adequately resolving inflammation due to its instability. However, the conjugation of novokinin with PEG and BP helped to improve stability and sustain its anti-inflammatory action. The upregulation of ACE2/ACE, Ang1-7/Ang II, and AT2R/AT1R ratios by Novo Conj in arthritic rats confirms its role in maintaining the physiological balance of RAS components for the resolution of inflammation. In addition, the enhanced production of anti-inflammatory ArA metabolites, EETs, and DiHTs was due to prolonged AT2R activation by Novo Conj, which further promotes its anti-inflammatory effects. Thus, bone-targeted delivery of Novo Conj supported by HA binding affinity data (Figure 2) and bone tissue distribution of similar conjugate (Ang 1-7 conjugate) [27] offers efficacious alternative therapy for the management of RA and could be utilized for various RAS disorders.

In summary, AT2R stands out as an attractive receptor-targeting approach for managing RA due to its influence on the RAS and ArA pathways. However, the AT2R peptide agonist, novokinin, falls short of efficiently resolving inflammation due to instability. The conjugation of novokinin with PEG and BP helps to improve stability and sustain its anti-inflammatory action. The upregulation of ACE2/ACE, Ang1-7/Ang II, and AT2R/AT1R ratios by Novo Conj in arthritic rats confirmed its role in maintaining the physiological balance of RAS components for the resolution of inflammation. Similar effects of Novo Conj on restoring the balance between anti- and pro-inflammatory metabolites of ArA are in concert with the RAS. The dynamic cross-talk between these two pathways and the Novo Conj effect on restoring their physiological homeostasis is very significant and indicative of its feasibility in controlling inflammatory conditions. Hence, the bone-targeted delivery of novokinin offers an effective alternative therapy for the modulation of RA and could be further utilized for various RAS disorders.

## Figures and Tables

**Figure 1 pharmaceutics-14-01681-f001:**
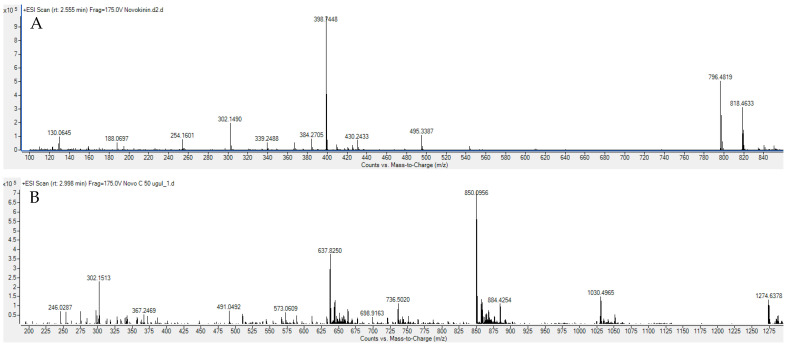
Extracted ion chromatogram (EIC) of (**A**) novokinin and (**B**) Novo Conj through LC/Q-TOF.

**Figure 2 pharmaceutics-14-01681-f002:**
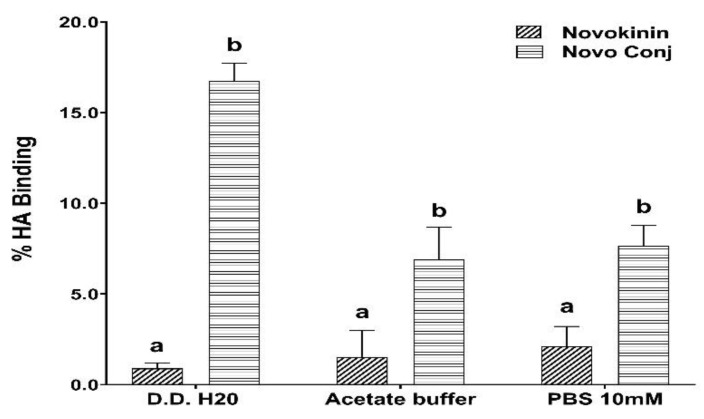
Novokinin conjugate (Novo Conj) shows higher hydroxyapatite (HA) binding affinity than novokinin. Samples were run in triplicate. Data labeled with different letters (a or b) indicate a statistical difference between groups where *p* < 0.05.

**Figure 3 pharmaceutics-14-01681-f003:**
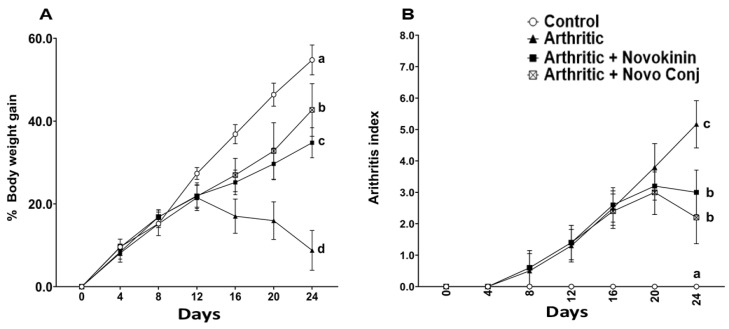
Treatment with novokinin conjugate (Novo Conj) restored body weight (**A**) and decreased arthritis index (AI) score (**B**). Data labeled with different letters (a, b, c, or d) indicate a statistical difference between groups where *p* < 0.05.

**Figure 4 pharmaceutics-14-01681-f004:**
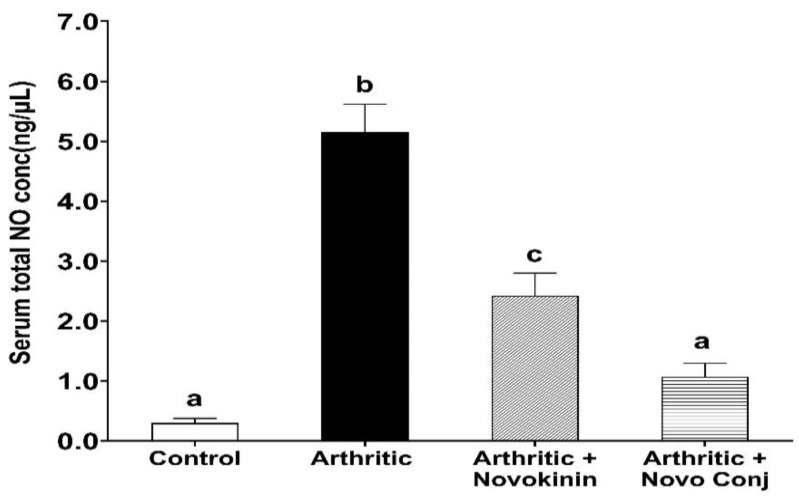
Treatment with novokinin conjugate (Novo Conj) reduced arthritis (AIA) induced elevated serum level of nitric oxide (NO). Mean ± SD (*n* = 5 or 6 rats per group). Data labeled with different letters (a, b, or c) indicate a statistical difference between groups where *p* < 0.05.

**Figure 5 pharmaceutics-14-01681-f005:**
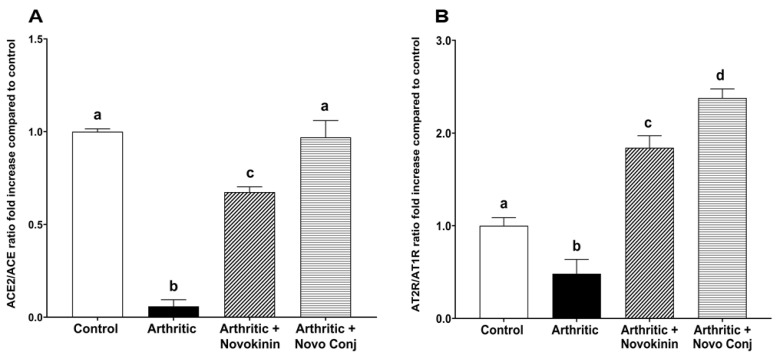
Novokinin conjugate (Novo Conj) increased mRNA expression of (**A**) the ratio of angiotensin-converting enzyme 2 (ACE2)/angiotensin-converting enzyme (ACE) and (**B**) the ratio of angiotensin type II receptor (AT2R)/angiotensin type I receptor (AT1R) in cardiac tissues. Mean ± SD (*n* = 5 or 6). Triplicates samples were run. Data labeled with different letters (a, b, c, or d) indicate a statistical difference between groups where *p* < 0.05.

**Figure 6 pharmaceutics-14-01681-f006:**
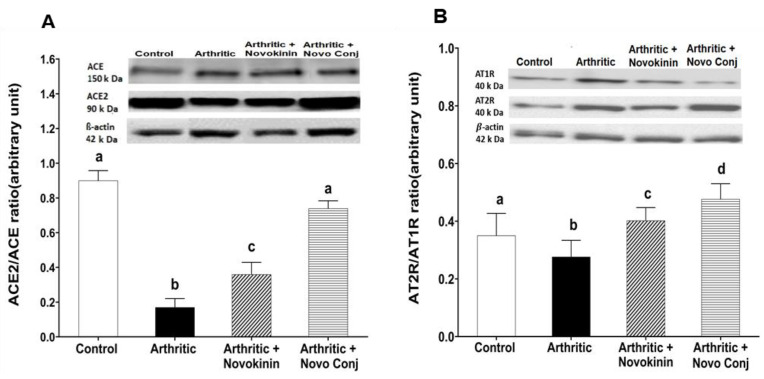
Novokinin conjugate (Novo Conj) increased protein expression of (**A**) angiotensin-converting enzyme 2 (ACE2)/angiotensin-converting enzyme (ACE1) and (**B**) angiotensin type II receptor (AT2R)/angiotensin type I receptor (AT1R) ratio in cardiac tissues. Mean ± standard deviation of the mean (*n* = 5 or 6). Triplicates samples were run. Data labeled with different letters (a, b, c, or d) indicate a statistical difference between groups where *p* < 0.05.

**Figure 7 pharmaceutics-14-01681-f007:**
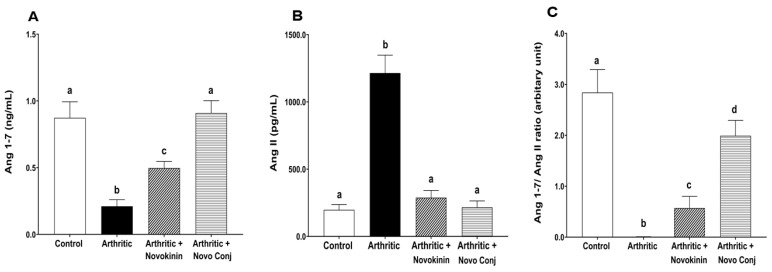
Novokinin and novokinin conjugate (Novo Conj) significantly increased angiotensin 1-7 (Ang 1-7) (**A**), decreased angiotensin II (Ang II) levels (**B**), and increased the Ang1-7/Ang II ratio (**C**) in plasma of adjuvant-induced arthritis (AIA) rats. Mean ± SD (*n* = 5 or 6). Triplicates samples were run. Data labeled with different letters (a, b, c, or d) indicate a statistical difference between groups where *p* < 0.05.

**Figure 8 pharmaceutics-14-01681-f008:**
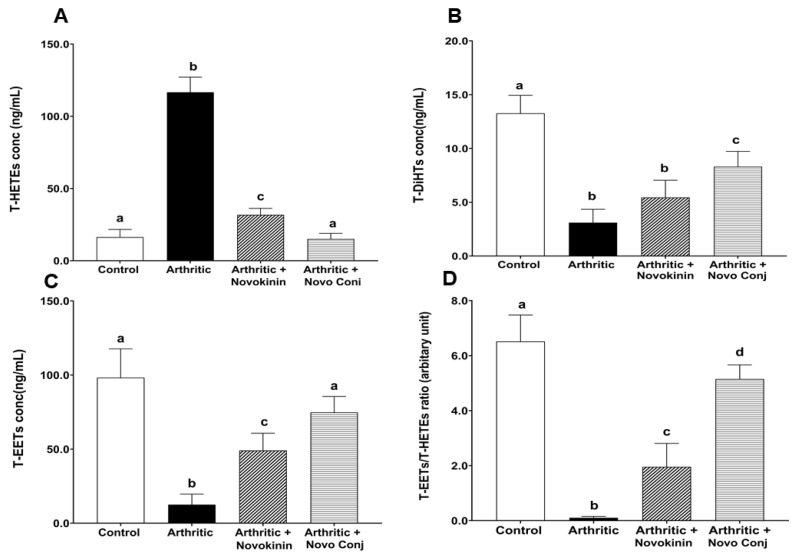
Impact of novokinin and novokinin conjugate (Novo Conj) treatment on total-HETEs (T-HETEs) (**A**), Total-DiHTs (T-DiHTs) (**B**), Total-EETs (T-EETs) (**C**), and the ratio of T-EETs/T-HETEs (**D**) arthritic rats plasma. Mean ± SD (*n* = 5 or 6). Data labeled with different letters (a, b, c, or d) indicate a statistical difference between groups where *p* < 0.05.

**Table 1 pharmaceutics-14-01681-t001:** Primer sequences of genes with the accession number and amplicon size.

Gene	Sequence	AN	AS	Ref
AT2R	5′-GGTCTGCTGGGATTGCCTTAATG-3′	NM_001385624.1	142	[29]
	5′-ACTTGGTCACGGGTAATTCTGTTC-3′			
AT1R	5′-GGAAACAGCTTGGTGGTGAT-3′	NM_008771594.3	171	[30]
	5′-ACTAGGTGATTGCCGAAGG-3′			
ACE2	5′-ACCCTTCTTACATCAGCCCTACTG-3′	NM_001012006.2	74	[31]
	5′-TGTCCAAAACTACCCCACATAT-3′			
ACE	5′-TTTGCTACACAAATGGCACTTGT-3′	NM_012544.1	67	[32]
	5′-CGGGACGTGGCCATTATATT-3′			
GAPDH	5′-CCTGCACCACCAACTGCTTA-3′	NM-017008.4	95	[33]
	5′-AGTGATGGCATGGACTGTGG-3′			

AT2R—angiotensin type II receptor; AT1Ra—angiotensin type I receptor a; ACE2—angiotensin type converting enzyme 2; ACE—angiotensin type converting enzyme; GAPDH— glyceraldehyde-3-phosphate dehydrogenase; AN—accession number; AS—amplicon size.

**Table 2 pharmaceutics-14-01681-t002:** Inter- and intra-day precision and accuracy of angiotensin peptides.

Concentration	Intra-Day		Inter-Day	
	CV (%)	Accuracy (%)	CV (%)	Accuracy (%)
**Ang 1-7 (ng/mL)**				
0.31	9.39	105.61	6.64	102.74
1.25	3.42	95.11	1.29	108.09
5	1.34	100.78	2.17	98.95
**Ang II (pg/mL)**				
200	5.05	102.94	8.45	96.81
800	2.39	102.52	5.69	100.95
3200	1.076	99.12	0.95	101.03

Ang 1-7, angiotensin 1-7; Ang II, angiotensin II; CV, coefficient of variance.

**Table 3 pharmaceutics-14-01681-t003:** Compound parameters for ArA metabolites and the deuterated IS with MRM in negative electrospray ionization mode are given below.

Analyte	Q1 *m*/*z*	Q3 *m*/*z*	DPV	CE	CXP	EP
5,6-EET	319	191	−30	−14	−11	−10
8,9-EET	319	155	−30	−14	−11	−10
11,12-EET	319	208	−120	−18	−11	−10
11,12-EET	319	167	−120	−18	−11	−10
14,15-EET	319	219	−105	−16	−13	−10
5,6-DiHT	337	145	−115	−22	−11	−10
5,6-DiHT	337	145	−115	−22	−11	−10
8,9-DiHT	337	127	−105	−26	−11	−10
11,12-DiHT	337	167	−115	−24	−19	−10
14,15-DiHT	337	207	−40	−24	−13	−10
19-HETE	319	231	−125	−20	−12	−10
20-HETE	319	245	−95	−22	−15	−10

EET, epoxyeicosatrienoic acid; HETE, hydroxyeicosatetraenoic acid; DiHT, dihydroxyeicosatrienoic acid; DP, declustering potential; CE, collision energy; CXP, collision cell exit potential; EP, entrance potential.

**Table 4 pharmaceutics-14-01681-t004:** Inter- and intra-day precision and accuracy of ArA metabolites.

Concentration	Intra-Day		Inter-Day		
	CV (%)	Accuracy (%)	CV (%)	Accuracy (%)	LLOQ
5,6-EET (ng/mL)					0.07 ng/mL
0.62	8.76	101.883	4.53	107.73	
5	2.12	101.46	3.76	103.41	
40	7.23	98.27	6.36	99.30	
160	0.73	100.26	0.68	100.06	
8,9-EET (ng/mL)					0.28 ng/mL
0.62	3.13	106.12	7.74	101.86	
5	6.51	99.96	6.84	100.53	
40	4.89	100.66	4.32	99.86	
160	0.34	100.22	0.22	100.56	
11,12-EET (ng/mL)					0.45 ng/mL
0.62	3.57	103.76	7.62	97.68	
5	0.41	102.64	9.76	98.04	
40	1.92	98.91	2.31	96.67	
160	0.17	100.45	1.71	101.78	
14,15-EET (ng/mL)					0.31 ng/mL
0.62	7.54	100.26	6.69	96.53	
5	6.89	96.53	7.58	103.06	
40	6.55	99.525	5.74	101.35	
160	2.28	101.64	1.12	102.61	
5,6-DiHT (ng/mL)					0.14 ng/mL
0.62	5.97	101.33	9.37	97.06	
5	5.11	107.86	9.55	99.60	
40	2.95	103.15	5.85	104.37	
160	0.62	102.29	1.34	101.81	
8,9-DiHT (ng/mL)					0.01 ng/mL
0.62	9.74	101.33	9.35	100.26	
5	9.06	105.33	5.18	102.01	
40	3.06	102.28	3.11	100.23	
160	1.06	101.60	1.81	101.37	
11,12-DiHT (ng/mL)					0.20 ng/mL
0.62	5.00	104.80	7.92	101.54	
5	3.22	110.26	2.25	105.86	
40	2.73	106.32	2.43	103.44	
160	2.00	100.30	1.81	101.21	
14,15-DiHT (ng/mL)					0.12 ng/mL
0.62	4.76	99.62	3.91	103.78	
5	3.19	105.33	2.45	102.2	
40	2.17	103.44	4.46	101.95	
160	0.31	101.73	1.69	102.13	
19-HETE (ng/mL)					0.18 ng/mL
0.62	4.33	96.48	9.61	102.4	
5	5.49	110.40	4.23	114.6	
40	1.82	105.29	3.67	107.14	
160	2.81	102.69	2.26	102.27	
20-HETE (ng/mL)					0.25 ng/mL
0.62	4.87	105.51	3.38	106.56	
5	5.04	113.07	3.06	109.73	
40	2.91	106.89	1.87	106.87	
160	1.10	101.64	1.46	102.83	

EET, epoxyeicosatrienoic acid; HETE, hydroxyeicosatetraenoic acid; DiHT, dihydroxyeicosatrienoic acid; LLOQ, lower limit of quantification.

**Table 5 pharmaceutics-14-01681-t005:** Percent change in paw and joints diameter in different treatment groups on day 24 compared to day 0.

	% Increase in Paw Diameter		% Increase in Joint Diameter	
Group	L. Hind	R. Hind	L. Hind	R. Hind
Control	4.6(3.6) ^a^	4.3(2.6) ^a^	3.6(3.7) ^a^	5.8(3.2) ^a^
Arthritic	21.7(4.9) ^b^	23.7(4.9) ^b^	25.7(4.8) ^b^	24.9(5.7) ^b^
Arthritic + Novokinin	16.9(4.9) ^b^	15.8(3.9) ^b^	18.7(3.4) ^c^	18.1(6.3) ^b^
Arthritic + Novo Conj	13.8(5.9) ^c^	11.8(4.9) ^c^	10.4(3.5) ^d^	12.49(2.7) ^c^

Novo Conj, novokinin conjugate; L. Hind, left hind; R. Hindm right hind. Mean ± SD (*n* = 5 or 6). Data labeled with different letters (a, b, c, or d) indicate a statistical difference between groups where *p* < 0.05.

**Table 6 pharmaceutics-14-01681-t006:** Significant linear correlations between RAS peptides and ArA metabolites.

	Ang 1-7		Ang II	
Plasma ArA Metabolites	r	*p* Value	r	*p* Value
11,12- EET	0.7188	0.0002	−0.5613	0.0066
14,15-EET	0.7245	0.0001	−0.6809	0.0005
T-EETs	0.7110	0.0002	−0.5887	0.0039
19-HETE	−0.7001	0.0003	0.6209	0.0021
20-HETE	−0.7379	<0.0001	0.7518	<0.0001
T-HETEs	−0.7381	<0.0001	0.7474	<0.0001
T-EETs/T-HETEs	0.6505	0.0010	−0.5382	0.0098
5,6-DiHT	0.5525	0.0077	−0.4943	0.0194
8,9-DiHT	0.6846	0.0004	−0.5679	0.0058
11,12-DiHT	0.4643	0.0295	−0.5157	0.0140
T-DiHTs	0.4674	0.0283	−0.5619	0.0065
T-DiHTs/T-HETEs	0.4693	0.0276	−0.4631	0.0301

EET, epoxyeicosatrienoic acid; T-EETs, Total-EETs; HETE, hydroxyeicosatetraenoic acid; T-HETEs, Total-HETE; DiHT, dihydroxyeicosatrienoic acid; T-DiHTs, Total-DiHTs.

## Data Availability

The data presented in this study are available on request from the corresponding author.

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
