# Peer review of "Bone-Targeted Delivery of Novokinin as an Alternative Treatment Option for Rheumatoid Arthritis"

_pharmaceutics, 2022, doi:10.3390/pharmaceutics14081681_

Round 1
Reviewer 1 Report
The manuscript entitled "Bone-Targeted Delivery of Novokinin as Alternative Treatment Options for Rheumatoid Arthritis" describes the results of an investigation that aims at the assessment of efficacy and mechanisms of action of a novel bone targeting Novokinin Conjugate (Novo Conj) in rats with adjuvant-induced arthritis.
In general, the work is properly designed and performed, and the manuscript is easy to read and follow. Most of the methods and results are explained thoroughly and in detail. However, the reviewer has some points related to the methodology, conclusions, and the results of statistical analysis:
1. Why did the authors select a model of adjuvant-induced arthritis as a model of rheumatoid arthritis (RA)? According to the literature, the most relevant model of RA in rats is a collagen-induced arthritis model. Please provide rationale for this selection and discuss briefly in the discussion section.
2. Why the numbers of rats were different in the study and control groups (5 vs. 6)? Please provide explanation.
3. What was the percent of immunized animals that developed the symptoms of RA? Please specify in the method section. What were the inclusion/exclusion criteria for rats in the study?
4. How were the doses of Novokinin and Novo Conj selected? Please justify and discuss briefly in the Discussion section. (line 144 – provide the route of administration.
5. line 162 - What was the LLOQ of NO assay? Please provide this information in the methods section.
6. Line 300: Were the data checked for normal distribution and homogeneity of variance before parametric statistical tests were used? Please specify in the methods section and provide the names of statistical procedures.
7. Arthritis index (AI) is not a continuous variable, so the mean values of multiple groups should be compared using a non-parametric statistical test, such as Kruskal–Wallis analysis of variance.
8. What is the exact meaning of the letters ‘a’, ‘b’, ‘c’, and ‘d’ in Figures? Please provide detailed explanation since this may be confusing to the reader. What were the exact p values for each comparison? Please provide this in Figures or in separate tables.
9. line 398 - The authors state that they found correlations between ArA metabolites levels and RAS components. Was it a linear correlation? What software was used to perform this analysis? Please specify. Please provide the graphs with fittings (in the supplementary materials). In this analysis, the authors should use orthogonal regression to account for errors in observations on both the x- and the y- axis.
10. Please provide more information related to LC-MS/MS methods’ development. Precision, accuracy range, and LOQ should be provided in the methods section.
11. The authors suggest in the conclusions (lines 517, 531) and throughout the manuscript that Novo Conj allows for bone-targeted delivery of Novokinin. Did the authors assess the distribution of Novo Conj to bones? If it was not assessed, the authors cannot conclude that the effects observed in this investigation are due to accumulation/distribution of Novo Conj in bones. In this situation, the title of the manuscript needs to be changed and phrases suggesting bone-targeted delivery of Novokinin needs to be removed.
Author Response
Dear Respected Reviewer,
We are very thankful that you found our manuscript adequately designed and performed, and it is easy to read and follow. We are welcoming the constructive comments and addressing them point-by-point below:
- Why did the authors select a model of adjuvant-induced arthritis as a model of rheumatoid arthritis (RA)? According to the literature, the most relevant model of RA in rats is a collagen-induced arthritis model. Please provide rationale for this selection and discuss briefly in the discussion section.
Answer: We agree with the reviewer’s comment that collagen-induced arthritis is the most relevant model; however, a lot of literature indicates that the AIA model is indeed an appropriate model that resembles human RA. We are interested in investigating the cardiovascular protection of Novo conj through RAS and ArA pathway, therefor based on some literature that claims that Adjuvant-induced arthritis is a relevant model to mimic coronary and myocardial impairments in rheumatoid arthritis; therefore, we chose to use Adjuvant-induced arthritis model as an appropriate model to mimic coronary and myocardial impairments in rheumatoid arthritis. The following references ( 25&26) were added to justify our choice of the animal model.
-Bordy R. et al. Adjuvant-induced arthritis is a relevant model to mimic coronary and myocardial impairments in rheumatoid arthritis. Jt. Bone Spine. 2021;88:105069. doi: 10.1016/j.jbspin.2020.09.001
- Bendele A, Mccomb J, Gould T, Mcabee T, Sennello G, Chlipala E, Guy M. 1999. Animal models of arthritis: Relevance to human disease. Toxicol Pathol 27:134–142.
- Why the numbers of rats were different in the study and control groups (5 vs. 6)? Please provide explanation.
Answer: One extra rat in control and the arthritic group was added to use their plasma and tissue samples for developing different experimental methods.
- What was the percent of immunized animals that developed the symptoms of RA? Please specify in the method section. What were the inclusion/exclusion criteria for rats in the study?
Answer: The percentage of arthritis development was 100%, with some variation in the onset of signs and symptoms. The inclusion criteria for the rats were to observe at least one paw or joint swelling; any rats who developed AI> 5 were excluded and euthanized. This statement has been added to the manuscript.
- How were the doses of Novokinin and Novo Conj selected? Please justify and discuss briefly in the Discussion section. (line 144 – provide the route of administration.
Answer: It has been reported that Novokinin was used in a wide range of 0.1 mg/kg intravenously to 30–100 mg/kg orally (-PMID: 18207609 DOI: 10.1016/j.peptides.2007.11.017). In this study, animals tolerated the Novo Conj’s administered dose well. We chose this as a pilot study to test its anti-inflammatory effect after subcutaneous injection. Later, we will conduct a dose-response study on a wide range to find a more effective and safe dose.
- line 162 - What was the LLOQ of NO assay? Please provide this information in the methods section.
Answer: Based on the kit’s manufacturer pamphlet, the LLOQ was about 82.5 picogram/μL, and our data were way above that. This statement has been added to the method section.
- Line 300: Were the data checked for normal distribution and homogeneity of variance before parametric statistical tests were used? Please specify in the methods section and provide the names of statistical procedures.
Answer: All data were analyzed for normal distribution and homogeneity of variance before proceeding with the parametric statistical tests.
- Arthritis index (AI) is not a continuous variable, so the mean values of multiple groups should be compared using a non-parametric statistical test, such as Kruskal–Wallis analysis of variance.
Answer: As a non-parametric statistical test, the mean values of the AI as a continuous variable were analyzed using the Kruskal-Wallis analysis of variance followed by the Mann-Whitney U-test to determine the significant difference between groups.
- What is the exact meaning of the letters’ a’, ‘b’, ‘c’, and ‘d’ in Figures? Please provide detailed explanation since this may be confusing to the reader. What were the exact p values for each comparison? Please provide this in Figures or in separate tables.
Answer: Data labeled with different letters (a, b, c, or d) indicate a statistical difference between groups where p <0.05.
- line 398 - The authors state that they found correlations between ArA metabolites levels and RAS components. Was it a linear correlation? What software was used to perform this analysis? Please specify. Please provide the graphs with fittings (in the supplementary materials). In this analysis, the authors should use orthogonal regression to account for errors in observations on both the x- and the y- axis.
Answer: The correlation between ArA metabolites levels and RAS components was linear Pearson correlation and performed using GraphPad software. To address the reviewer’s comment, the orthogonal regression has been conducted using Minitab software, which was mentioned in the statistical analysis section, and the output has been reported in supplementary data.
- Please provide more information related to LC-MS/MS methods’ development. Precision, accuracy range, and LOQ should be provided in the methods section.
The LC-MS/MS methods were adopted from previously published methods and used after validating their robustness. A brief set of validation data is included.
- The authors suggest in the conclusions (lines 517, 531) and throughout the manuscript that Novo Conj allows for bone-targeted delivery of Novokinin. Did the authors assess the distribution of Novo Conj to bones? If it was not assessed, the authors cannot conclude that the effects observed in this investigation are due to accumulation/distribution of Novo Conj in bones. In this situation, the title of the manuscript needs to be changed and phrases suggesting bone-targeted delivery of Novokinin needs to be removed.
Answer: We did study the tissue distribution of similar conjugate (Ang Conj) with similar peptide structure, linker, and bone targeting moiety. The in vitro and in vivo study result indicates that the conjugate accumulates on the bone significantly. Similarly, the in vitro result of HA binding of Novo conjugate (Figure 2) replicates what we saw with Ang Conj (Ref #27). Therefore, it is fair to consider that Novo Conj also targets the bone similarly.
Reviewer 2 Report
The manuscript entitled “Bone-Targeted Delivery of Novokinin as Alternative Treatment Options for Rheumatoid Arthritis” addresses the beneficial effect of novokinin conjugate in vivo in adjuvant-induced arthritis in rats. The authors also explored the associated molecular mechanisms. Initially, the authors have proven that the novokinin conjugate attenuates the clinical signs of arthritis. These favorable actions were established by dampening the expression of the pro-inflammatory renin-angiotensin system (RAS) components and arachidonic acid metabolites in cardiac tissue and plasma. The current findings are interesting.
Comments:
1) In the experimental design of the animal study (section 2.2.1. Induction and Assessment of Experimental AA), why did not the authors incorporate an additional experimental group (control + novo conj.). This group may reveal any potential toxicity of the test agent in rats at the indicated dose.
2) What is the LD50 for novokinin or novo conj. in rats? Is the used dose safe?
3) In the statistical analysis section, did the authors check data normality and homogeneity before proceeding to one-way ANOVA?
4) In section 2.2.1. (Induction and Assessment of Experimental AA), how did the authors decide on the dose of the novokinin and novokinin conjugate in rats? How is the dose relevant to the human dose using the Human effective dose (HED) formula= animal dose x animal Km/ human Km (Nair AB, Jacob S. A simple practice guide for dose conversion between animals and humans. J Basic Clin Pharm. 2016 Mar;7(2):27-31). Authors are advised to address this point and add the answers to the comment in section 2.2.1. Please also provide proper citations for selecting such doses.
5) In q-PCR: The authors are advised to add the gene accession number, amplicon size for all target genes. Please, add these data in the material and methods section
6) The author should mention the amount of RNA used for the synthesis of cDNA and the amount of cDNA used for qRT-PCR. Please, add these data in the material and methods section
7) The qRT-PCR is missing the biological repeat (how many samples were used per experimental group). Moreover, did the authors check the RNA quality with A260/280, and perform an RT negative control to ensure no DNA contamination in the RNA extraction? Please, add these data in the material and methods section.
8) To make all figure legends stand-alone, authors are advised to add the full name of the used abbreviations at the end of each legend.
9) Given the fact that the arthritic index scores are non-parametric data, ANOVA analysis is not an appropriate test. The authors are advised to express the non-parametric data as medians and to analyze the data using Kruskal-Wallis analysis of variance. When statistical significance is obtained, the rank-based Mann–Whitney U-test can be used to compare the groups.
10) To avoid confusion of readers and to clarify the names of experimental groups, the authors are advised to rename the experimental groups described in section 2.2.1. and table 2, for example, as control, arthritic (instead of inflamed), arthritic + novokinin (instead of novokinin), and arthritic + Novo conj. (instead of Novo Conj). Please, address this issue in the entire manuscript.
11) In Figure 3, the authors are advised to label “total NO” as “serum total NO”. Why did not the authors investigate the levels of NO in paw tissue, In fact, since serum is a pool for all inflammatory events in the body and there is no guarantee that the serum inflammatory changes originated entirely from the paw, Serum NO cannot show exactly the paw/local NO concentration changes.
12) Likewise, the authors should have investigated the expression levels of Ang 1-7 and Ang II in paw tissue rather than in plasma.
13) In sections 3.2.3. and 3.2.4., why did the authors determine the protein expression of RAS components in the cardiac tissue. In fact, the aim of the present work is to investigate the effect of novokinin conjugate on arthritis rather than on the cardiovascular system. The authors should have investigated RAS protein expression in paw tissue which can directly describe the arthritis microenvironment.
14) In line 339, the authors state that “NO is a biomarker of inflammation that gets elevated in the AIA rats compared with the control rats”. The above statement is not accurate since NO is a marker of nitrosative stress rather than a direct marker of inflammation. Please, correct the issue.
15) None of the experiments addressed the targeted delivery of novokinin to bone tissue. Herein, bone targeting was not proven and thus the title of the current study needs to be modified accordingly.
Author Response
Dear Respected Reviewer,
We are very grateful that you considered our findings interesting. We value your positive comments by addressing them point-by-point as below:
Comments:
1) In the experimental design of the animal study (section 2.2.1. Induction and Assessment of Experimental AA), why did not the authors incorporate an additional experimental group (control + novo conj.). This group may reveal any potential toxicity of the test agent in rats at the indicated dose.
Answer: We appreciate the reviewer’s comment. This is a pilot study for evaluating the Novo conjugate efficacy. I hope that in a future study, we will investigate the possible toxicity over a wide range of doses.
2) What is the LD50 for novokinin or novo conj. in rats? Is the used dose safe?
There is no report of LD50 for Novokinin. In this study, animals tolerated the Novo Conj’s administered dose well.
3) In the statistical analysis section, did the authors check data normality and homogeneity before proceeding to one-way ANOVA?
Answer: All data were analyzed for normal distribution and homogeneity of variance before proceeding with the parametric statistical tests.
4) In section 2.2.1. (Induction and Assessment of Experimental AA), how did the authors decide on the dose of the novokinin and novokinin conjugate in rats? How is the dose relevant to the human dose using the Human effective dose (HED) formula= animal dose x animal Km/ human Km (Nair AB, Jacob S. A simple practice guide for dose conversion between animals and humans. J Basic Clin Pharm. 2016 Mar;7(2):27-31). Authors are advised to address this point and add the answers to the comment in section 2.2.1. Please also provide proper citations for selecting such doses.
Answer: No human study has been done on Novokinin. It has been reported that Novokinin was used in a wide range of 0.1 mg/kg intravenously to 30–100 mg/kg orally ( Ref #42&43). We chose this as a pilot study to test its anti-inflammatory effect after subcutaneous injection. Later, we will conduct a dose-response study on a wide range to find a more effective and safe dose.
5) In q-PCR: The authors are advised to add the gene accession number, amplicon size for all target genes. Please, add these data in the material and methods section
Answer: The suggested information has been added to the method section
6) The author should mention the amount of RNA used for the synthesis of cDNA and the amount of cDNA used for qRT-PCR. Please, add these data in the material and methods section
Answer: The suggested information has been added to the method section
7) The qRT-PCR is missing the biological repeat (how many samples were used per experimental group). Moreover, did the authors check the RNA quality with A260/280, and perform an RT negative control to ensure no DNA contamination in the RNA extraction? Please, add these data in the material and methods section.
Answer: As suggested, the biological repeats have been added to the figure legends. The RNA quality and DNA contamination were determined using a Qubit 2.0 fluorometer. Furthermore, the RNA extraction kit described in the methods section was followed as guided by the manufacturer’s instructions. In the steps, both gDNA was eliminated with a column directed towards gDNA and using DNase. Lastly, the primers used in our study were derived from previously published studies. Most of the primer combinations are intron spanning, thus eliminating the chance of gDNA at another level. With all of these procedures, gDNA is unlikely to contribute to the cycle threshold values seen in our qPCR.
8) To make all figure legends stand-alone, authors are advised to add the full name of the used abbreviations at the end of each legend.
Answer: All the abbreviation was spelled out in each legend
9) Given the fact that the arthritic index scores are non-parametric data, ANOVA analysis is not an appropriate test. The authors are advised to express the non-parametric data as medians and to analyze the data using Kruskal-Wallis analysis of variance. When statistical significance is obtained, the rank-based Mann–Whitney U-test can be used to compare the groups.
Answer: As a non-parametric statistical test, the mean values of the AI as a continuous variable were analyzed using the Kruskal-Wallis analysis of variance followed by the Mann-Whitney U-test to determine the significant difference between groups.
10) To avoid confusion of readers and to clarify the names of experimental groups, the authors are advised to rename the experimental groups described in section 2.2.1. and table 2, for example, as control, arthritic (instead of inflamed), arthritic + novokinin (instead of novokinin), and arthritic + Novo conj. (instead of Novo Conj). Please, address this issue in the entire manuscript.
Answer: All suggested changes were applied.
11) In Figure 3, the authors are advised to label “total NO” as “serum total NO”. Why did not the authors investigate the levels of NO in paw tissue, In fact, since serum is a pool for all inflammatory events in the body and there is no guarantee that the serum inflammatory changes originated entirely from the paw, Serum NO cannot show exactly the paw/local NO concentration changes.
Answer: We were interested in investigating the cardiovascular protection of Novo conj through the RAS and ArA pathway; therefore, we measured serum levels of NO. In future studies, we will explore the local level as well.
12) Likewise, the authors should have investigated the expression levels of Ang 1-7 and Ang II in paw tissue rather than in plasma.
Answer: Please see the reply to comment #11.
13) In sections 3.2.3. and 3.2.4., why did the authors determine the protein expression of RAS components in the cardiac tissue. In fact, the aim of the present work is to investigate the effect of novokinin conjugate on arthritis rather than on the cardiovascular system. The authors should have investigated RAS protein expression in paw tissue which can directly describe the arthritis microenvironment.
Answer: We agree with the reviewer’s comment that we pursue further than the cardiovascular system and include other tissues. Due to the high workload, we did limit our experiment to serum and plasma cardiac tissue for now. We have collected all other tissues and will analyze them later.
14) In line 339, the authors state that “NO is a biomarker of inflammation that gets elevated in the AIA rats compared with the control rats”. The above statement is not accurate since NO is a marker of nitrosative stress rather than a direct marker of inflammation. Please, correct the issue.
Answer: We agree with the comment. Accordingly, the statement is corrected to “NO is a marker of nitrosative stress, and it directly participates in the pathogenesis of RA as reported in several studies. So we measured serum total NO as an indication of resolution of inflammation by the treatment of Novo Conj.”
15) None of the experiments addressed the targeted delivery of novokinin to bone tissue. Herein, bone targeting was not proven and thus the title of the current study needs to be modified accordingly.
Answer: We did study the tissue distribution of similar conjugate (Ang Conj) with similar peptide structure, linker, and bone targeting moiety. The in vitro and in vivo study result indicates that the conjugate accumulates on the bone significantly. Similarly, the in vitro result of HA binding of Novo conjugate (Figure 2) replicates what we saw with Ang Conj (Ref 27). Therefore, it is fair to consider that Novo Conj also targets the bone similarly.
Round 2
Reviewer 1 Report
The authors have satisfactorily addressed most of the comments, and this manuscript has been substantially improved.
Reviewer 2 Report
The authors have adequately addressed raised comments.